# Updated Mechanisms of GCN5—The Monkey King of the Plant Kingdom in Plant Development and Resistance to Abiotic Stresses

**DOI:** 10.3390/cells10050979

**Published:** 2021-04-22

**Authors:** Lei Gan, Zhenzhen Wei, Zuoren Yang, Fuguang Li, Zhi Wang

**Affiliations:** 1Zhengzhou Research Base, State Key Laboratory of Cotton Biology, Zhengzhou University, Zhengzhou 450001, China; ganleishengwu@163.com (L.G.); wzz19920315@163.com (Z.W.); yangzuoren4012@163.com (Z.Y.); aylifug@caas.cn (F.L.); 2State Key Laboratory of Cotton Biology, Key Laboratory of Biological and Genetic Breeding of Cotton, Institute of Cotton Research, Chinese Academy of Agricultural Sciences, Anyang 455000, China

**Keywords:** histone modification, GCN5, ADA2b, organ development, trichome, signaling pathways, abiotic stress

## Abstract

Histone modifications are the main epigenetic mechanisms that regulate gene expression, chromatin structure, and plant development, among which histone acetylation is one of the most important and studied epigenetic modifications. Histone acetylation is believed to enhance DNA access and promote transcription. GENERAL CONTROL NON-REPRESSIBLE 5 (GCN5), a well-known enzymatic protein responsible for the lysine acetylation of histone H3 and H4, is a universal and crucial histone acetyltransferase involved in gene transcription and plant development. Many studies have found that GCN5 plays important roles in the different development stages of *Arabidopsis*. In terms of exogenous stress conditions, GCN5 is also involved in the responses to heat stress, cold stress, and nutrient element deficiency by regulating the related gene expression to maintain the homeostasis of some key metabolites (e.g., cellulose) or ions (e.g., phosphate, iron); in addition, GCN5 is involved in the phytohormone pathways such as ethylene, auxin, and salicylic acid to play various roles during the plant lifecycle. Some of the pathways involved by GCN5 also interwind to regulate specific physiological processes or developmental stages. Here, interactions between various developmental events and stress-resistant pathways mediated by GCN5 are comprehensively addressed and the underlying mechanisms are discussed in the plant. Studies with some interacting factors such as ADA2b provided valuable information for the complicated histone acetylation mechanisms. We also suggest the future focuses for GCN5 functions and mechanisms such as functions in seed development/germination stages, exploration of novel interaction factors, identification of more protein substrates, and application of advanced biotechnology-CRISPR in crop genetic improvement, which would be helpful for the complete illumination of roles and mechanisms of GCN5.

## 1. Introduction

Epigenetics play important roles in eukaryotes development, including reversible modifications on DNA and chromatin, post-transcriptional and post-translational embellishment, amongst which histone modifications, DNA and RNA methylations, and the regulation of non-coding RNA are the three pillars in epigenetic studies. At present, the known main modifications of histone include acetylation, methylation, phosphorylation, ubiquitylation, sumoylation, ribosylation, etc. [1], which play important regulatory roles in the processes of DNA replication, gene transcription, and chromatin aggregation, and so on. Among the kinds of histone modifications, histone acetylation is well-studied and it mainly occurs at lysine residues (K) on histones H3 and H4 to regulate gene transcription and chromatin remodeling. Physically, histone acetylation is the transfer of acetyl groups from acetyl-CoA to histone lysine residues, which neutralizes the positive charge of the histone tail and reduces the affinity of the histone tail to negatively charged DNA; thus, it promotes the binding of transcription factors to DNA and, in turn, functions in transcription activation and DNA damage repair [2]. Furthermore, interactions between different histone modifications may play important roles in providing signals for the recruitment of specific chromatin-related proteins, leading to changes in chromatin status and transcriptional regulation [3]. In conclusion, histone acetylation is involved in different signaling and metabolic pathways to exert versatile regulation in plant growth and development as well as resistance to abiotic stresses.

In eukaryotes, histone acetylation maintains dynamic balance by two types of inverse enzymatic reactions mediated by histone acetyltransferase (HAT) and histone deacetylase (HDAC), respectively. HATs are divided into four categories in all plants: (1) HAG of the GNAT (GCN5-related N-terminal acetyltransferases) superfamily; (2) HAM of the MYST superfamily; (3) HAC of the CREB-binding protein (CBP) family; (4) HAF of the TATA-binding protein-associated factor (TAFII250) family [4]. GENERAL CONTROL NON-REPRESSIBLE 5 (GCN5/HAG1), as a key catalytic component of multiple acetyltransferase complexes, is the most studied and functional histone acetyltransferase in eukaryote. The GCN5 protein was first identified as a universal transcriptional co-activator by genetic studies in yeast [5] and plays an important role in gene expression and amino acid synthesis in yeast, and mutations of *GCN5* can lead to increasing or reducing a large number of genes expression [5]. In *Tetrahymena*, histone acetyltransferase activity was identified in the homologous proteins of GCN5 for the first time, and histone acetylation was linked to gene transcriptional regulation [6]. In *Arabidopsis*, thirteen *HAT* genes have been identified including four genes in the *HAG* subfamily *(HAG1-HAG3*, *MMC1)*, two genes in the *HAM* subfamily (*HAM1* and *HAM2*), five genes in the *HAC* subfamily (*HAC1*, *HAC2*, *HAC4*, *HAC5*, and *HAC12*), and two genes in the *HAF* subfamily (*HAF1* and *HAF2*) [3,7]. Some mutations in *GCN5* show pleiotropic development defects, including abnormal meristematic function, loss of apical dominance, dwarfism, shorter petals, and male flowers [8]. GCN5 also plays important roles in the resistance to abiotic stresses such as heat [9] and cold [10], as well as in maintaining the steady-state of iron [11] and phosphate in the cells [12]. Besides the higher plants, eight HATs were identified in the lower plant *Marchantia polymorpha* genome. The subcellular localization and expression analyses indicated that all MpHAT are functional proteins and involved in various signaling pathways, however, the functions of these orthologous genes including *MpGCN5* in plant development remain to be elucidated [13,14].

Ten years ago, the functions of GCN5 in plant development were reviewed [15]. Here, the updated roles and mechanisms of GCN5 in plant growth and development and resistance to abiotic stress are summarized in detail. The novel mechanisms by GCN5 are discussed and raised; the future focuses for GCN5 study in plant development and tolerance to abiotic stresses are also addressed.

## 2. Roles and Mechanisms of GCN5 in Plant Growth and Development

### 2.1. Molecular Mechanisms of GCN5 Involved in Plant Vegetative Growth

During growth and development, plants undergo a series of phase transitions from the juvenile to adult vegetative phase, then to the reproductive phase. The vegetative phase transition from juvenile to adult is controlled by a regulatory module consisting of conserved microRNAs of the *miR156* family and their target transcription factors SQUAMOSA PROMOTER BINDING PROTEIN LIKEs (SPLs) in *Arabidopsis* and maize [16,17]. In *Arabidopsis*, *miR156* transcripts gradually decrease in seedlings with aging, which results in a gradual increase of the transcripts of the target gene *SPLs* and the growth transition [18,19]. GCN5 and transcriptional coactivator-ADA2b (also referred to as Proporz1/PRZ1) constituting Spt-Ada-Gcn5 Acetyltransferase (SAGA)-like histone acetyltransferase complex also plays role in the determination of the time of juvenile-to-adult phase transition through directly controlling the transcription of homologous *SPL3* and *SPL9* by histone acetylation [16]. Interestingly, another study showed that the *pri-miR156a* was decreased in *gcn5-2* [20], but the specific correlation between HAT machinery and the *miR156* pathway is ambiguous. Some genetic experiments could be applied to explore the relationship between histone acetylation associated with GCN5 and *miR156* in the vegetative phase transition. These results indicate that epigenetic mechanisms such as histone acetylation and microRNA regulation are crucial for the developmental transitions of the plant, among which GCN5 is a key component of the developmental signal leading to transcriptional activation of *SPLs* and developmental stages transition (Figure 1).

The photosynthesis system plays a vital role in plant development. *Arabidopsis* histone acetyltransferase TATA-binding protein-associated factor (TAF1/HAF2) is necessary for leaf greening and photoactive gene transcription, and the deletion of the *TAF1* gene can lead to developmental abnormalities [21]. Furthermore, GCN5 and histone deacetylase HD1 are also involved in light regulation with antagonistic effects. Mutant *gcn5* results in a longer hypocotyl and reduced photoinduced gene expression, while *HD1* mutation produces opposite effects and phenotypes. Interestingly, the double mutant *gcn5 hd1* recovers to normal light morphological phenotype; Chromatin immunoprecipitation (ChIP) data further revealed that GCN5 and HD1, acetylates and deacetylates the histones in the promoters of same genes *CHLOROPHYLL A/B-BINDING PROTEIN 2* (*CAB2*), *RIBULOSE BISPHOS-PHATE CARBOXYLASE SMALL CHAIN 1A* (*RBCS1A*), and *INDOLE-3-ACETIC ACID INDUCIBLE 3* (*IAA3*), respectively, to involve the photomorphogenesis [22], indicating that both of them play opposite roles to keep the dynamic balance of histone acetylation level in the photomorphogenesis. In contrast, the double mutant *gcn5 taf1* displays a severe defect of light-regulating gene expression, suggesting that GCN5 and TAF1 control the histone acetylation and transcription of target genes synergistically [22]. Some biochemical research showed that GCN5 preferentially enhances the acetylation of histones H3 and H4 concentrated in the core promoter region, while HD1 decreases the acetylation on both the core and more upstream promoter regions. GCN5 or TAF1 is responsible for the acetylation of H3K9, H3K27, H4K5, and H4K12 on target promoters, while the acetylation of H3K14 is only dependent on GCN5. Interestingly, GCN5 and TAF1 can exhibit a cumulative effect on the acetylation of H3K9. On the other hand, HD1 reduces acetylation at H3K9, H3K27, and H4K8 [22]. ELONGATED HYPOCOTYL 5 (HY5) is a key transcription factor involved in light-regulated gene transcription and photomorphogenesis. Studies have shown that GCN5 and TAF1 interact with HY5 through different approaches to regulate downstream gene expression [14,21]. These results suggest that light-regulated gene expression requires a fine acetylation homeostatic balance at specific histone lysine residues controlled by a module made of GCN5-TAF1-HD1 (Figure 1), which contributes to the plant photomorphogenesis. This dynamic equilibrium of histone acetylation network mediated by GCN5-TAF1-HD1 model is complicated and elaborate, while the underlying mechanisms of various histones and lysines modifications by different HATs or HDACs complex are still unclear.

### 2.2. Underlying Mechanisms of GCN5 Regulating Root Meristem

In *Arabidopsis*, the root growth is maintained by both tissue center and stationary center stem cells [23]. The stem cell ecological balance in roots is specified by two parallel transcription factor pathways related with PLETHORA (PLT) and SHORT ROOT (SHR)/SCARECROW (SCR)/RETINOBLASTOMA RELATED (RBR) [24,25,26,27]. The PLTs have been proved to be dose-dependent root system growth regulators [28]. A high level of PLT2 maintains stem cell status, the middle level of that promotes the amplification of cell division and defines the meristem region, while the low level of PLTs allows differentiation progress. GCN5 also regulates root meristem through mediating the transcription of key genes. Mutant *gcn5* shows defects in root quiescent center specification and root meristem differentiation [29]. Genetically, the overexpression of *PLT2* can restore the stem cell niche defects of *gcn5* mutants, indicating that GCN5 regulates the root tip stem cell niche and root growth by mediating *PLT1* and *PLT2* transcription expression [8,15,29]. ADA2b, together with GCN5, can increase transcripts of *PLT1* and *PLT2*, and adjust the stem cells meristem partition and cell expansion, however, GCN5 but not ADA2b affects the stem cell niche maintenance [29]. In cereals, crown roots are important for crops to achieve water and nutrition as the main constituent of the fibrous root system. In rice, *ADA2* and *GCN5* genes show high expression in the root meristem, and the WUSCHEL-related homeobox protein WOX11 recruits the ADA2-GCN5 module to activate downstream target genes (e.g., *OsPIN9*, *OsCSLF6*, *Os1BGLU*), which establishes the gene expression programs in crown root meristem cell division of rice [30]. Thus, it is speculated that there may be different SAGA-like complexes associated with GCN5 and/or ADA2b, and various downstream genes to involve root stem cell production and maintenance. A recent study also showed that several root-meristem genes *WOX5, WOX14, SCR, PLT1*, and *PLT2* are regulated by histone acetylation mediated by GCN5 in developing callus, then their transcription is activated, conferring regeneration potential to callus and *de novo* shoot meristem [31], indicating the conserved roles of GCN5 in both root and shoot meristems regulation.

### 2.3. Underlying Mechanisms of GCN5 Regulating Shoot Meristems in Reproductive Growth

Flowering is a sign that plants develop into the reproductive stage from the vegetative phase. The mutation of *GCN5* can not only influence vegetative development such as dwarfism, leaf folding and serration, loss of apical advantage but also cause flowering defects such as reduced fertility and aberrant inflorescence [32]. The flower is the most complex plant organ. *Arabidopsis* has a typical eudicot flower with four concentric whorls. The first-outer whorl is composed of four sepals, the second whorl with four petals, the third with six stamens—two short lateral and four long medial, and the fourth whorl with two congenitally fused carpels, which form the gynoecium, the female part of the flower [33]. The well-known functional genes *AGAMOUS (AG)* and *WUSCHEL (WUS)* are two key factors in the floral meristem and flower morphogenesis [34]. CLAVATA1 (CLV1) and GCN5 are involved in the regulation of apical-basal and mediolateral polarity of the *Arabidopsis* gynoecium. They regulate gynoecium morphogenesis by negatively mediating auxin biosynthesis and promoting auxin polar transport. Moreover, they also negatively regulate cytokinin signaling at the stigma and advance its accumulation in the carpel margin meristem. However, the potential mechanisms and interaction between different phytohormones regulated by CLV1/GCN5 are still unclear. At the molecular level, they regulate the key WUS-AG pathway synergistically and directly by histone acetylation modifications at the center of the gynoecium and function in flowering [33]. Moreover, *gcn5* shows defects in the number and arrangement of stamens at the later stage of flower organ development. The detailed study displayed that organs in whorl two were transformed into stamens abnormally, consequently producing supernumerary stamens, increasing the number of stamens of *gcn5*. The aberrant flowering results in the termination of inflorescence meristem, which is also realized through the regulation of the WUS-AG pathway by GCN5 [32]. Recent research also showed that mutant *gcn5-1* displays germ cell excessive proliferation of phenotype, and an abnormal structure of dysplasia surrounded by inflorescence meristem (Figure 1) [33]. From all above, it indicates that *GCN5* affects the development of almost all the floral organs including stamens and pistils through different phytohormone (e.g., auxin, cytokinin) pathways and direct target of WUS in *Arabidopsis*, although the detailed mechanisms such as the interaction among different phytohormones and AG-WUS factors are not very clear.

Collectively, *GCN5* is very important for the maintenance of the root and shoot stem cell niche and meristem functioning upstream of some key potency factors (e.g., WOXs, SCR, and PLTs) (Figure 1), however, the complicated pathways associated with GCN5 still leave many questions to answer in the plant-cell regeneration and pluripotency regulation of different plant species.

### 2.4. Mechanisms of GCN5 Mediating Trichome Development

Trichomes are hair-like protuberances on the epidermis of the different parts of the plants, and help plants to cope with environmental stresses such as heat, water, and UV light [35]. *Arabidopsis* trichomes are unicellular non-glandular trichomes with one stalk and two to four branches developed from epidermal cells [36]. *GLABRA1 (GL1), GL2*, and *GL3* are all well-known core control genes for trichome initiation. *Arabidopsis gcn5-1* shows increased trichomes density, decreased expression of *GL1, GL2*, and *GL3*, as well as decreased acetylation level of H3K14/K9 in the transcription start site (TSS) regions of these three genes. Consistently, when the *GCN5* gene is over-expressed in *Arabidopsis*, the density of leaf surface trichomes is decreased. In addition, ADA2b, the interactor of GCN5 also affects the trichome branches of Arabidopsis. Through genetic and molecular approaches, it showed that the GCN5-ADA2b complex regulates the initiation of *Arabidopsis* trichomes and influences expressions of some key genes by controlling acetylation levels of the target genomic H3K14/K9 (Figure 1) [37]. Endoreduplication, which leads to increased ploidy, is thought to be responsible for the formation of a normal trichome. The ploidy level of leaves is reduced in the *gcn5* mutant, while increased in the *ada2b* mutant, suggesting the different endoreduplication pathways regulated by GCN5 and ADA2b [38]. So, it seems that GCN5 and ADA2b are involved in various pathways including some key genes (e.g., *GL1*, *GL2*, *GL3*) and endoreduplication regulation to affect the trichome development, however, the correlation between these pathways is yet unclear. The *gcn5-1* and *ada2b-1* mutants with the Wassilewskija (WS) *Arabidopsis* background show decreased trichome branches, while *gcn5-6* with Columbia Arabidopsis background shows increased branches [39], indicating that the regulation mechanism of the trichome development is dependent on the genetic backgrounds partially.

### 2.5. Mechanisms of GCN5 Contributing to Stem Cuticular Wax Formation

The stem cuticular wax of plants is one of the important organs to protect plants from biological invasion and abiotic stress to maintain normal plant growth. Lots of candidate genes have been identified in the cuticular wax biosynthetic pathway, most of which encode enzymatic proteins, or work with enzymes in the very-long-chain fatty acids (VLCFA) biosynthesis and derivatization pathways [40]. *ECERIFERUM3* (*CER3*), a key candidate gene of stem cuticular wax synthesis, is a target gene of GCN5. Compared with the wild type, the acetylation level of H3K9/K14 in the *CER3* promoter region of the *gcn5-2* mutant is significantly reduced, which results in the downregulation of its transcript. Additionally, the over-expression of *CER3* restored the biosynthesis defect of the cuticular wax of the stem caused by the GCN5 mutation [41]. These studies indicate that *CER3* transcription is regulated directly by GCN5 through histone acetylation, which is partially involved in the cuticular wax synthesis (Figure 1). However, whether GCN5 can regulate other genes in the VLCFA pathway still need further research.

In summary, GCN5 plays important roles in the development of *Arabidopsis* from juvenile to adult processes, and from vegetative growth to reproductive growth with multifarious downstream factors (Figure 1). As an epigenetic factor, *GCN5* is involved in the regulation of plant organs development through participating in H3K9/K14 acetylation process on various downstream genes or promoters. The deletion and destruction of *GCN5* and *ADA2b* genes in *Arabidopsis* severely affect the development and growth of stems, roots, and leaves [8,32], which proposes the implicit and core roles of *GCN5* and *ADA2b* in the SAGA complex associated with plant development. More interestingly, GCN5 also regulates stem cell development and cell pluripotency by the WUS-AG pathway, which reinforces the importance of epigenetic modifications in eukaryote development and cell regeneration and provides new ideas and clues for the utilization of histone acetyltransferase such as GCN5 to improve the production of beneficial plant organs and favorite traits of crops. However, whether GCN5 plays some roles in seed-related traits (e.g., seed dormancy establishment, seed maturation, seed germination) has not been reported.

## 3. The Mechanisms of GCN5 in Plant Hormonal and Secondary Metabolic Pathways

### 3.1. The Mechanisms of GCN5 Involved in Crosstalk of Ethylene and Auxin Pathways

Ethylene is a type of gaseous plant hormone involved in many developmental processes (e.g., seed germination, seedling morphology, fruit maturation, and leaf senescence) and responses to environmental stresses including, biological stress, and abiotic stress tolerance [42]. Both GCN5 and CLV1 are involved in the regulation of expression of WUS and thus the size of the meristem [32,43]. Interestingly, the *clv1-1 gcn5-1* double mutant exhibits constitutive ethylene reaction indicating that CLAVATA and GCN5 show an effect in ethylene signaling synergistically and negatively [44] This interaction module of gene-ethylene (the gaseous molecule) is mediated by ETHYLENE INSENSITIVE 3/ EIN3-Like1 (EIN3/EIL1) transcription factors. EIN3, a key transcription factor, is required to regulate the ethylene-induced gene expression and ethylene signaling through initiating downstream transcriptional cascades. In the *gcn5-1* mutant, it showed the reduced H3 acetylation in promoters of *ERF1* and *EBF2*, two direct targets of EIN3/EIL1, which was restored to similar to or even higher level in the double mutant *clv1-1 gcn5-1* than those in the wild type. Further, the expression levels of *ERF1* and *EBF2* were slightly increased after ethylene treatment in *clv1-1 gcn5-1* [44,45,46]. So, CLV signaling suppresses GCN5 action on H3 acetylation of *ERF1* and *EBF2* loci, and histone acetylation associated with GCN5 is the partial upstream activator of *ERF1* and *EBF2*. Intensive research indicated that the restored H3 acetylation in *clv1-1 gcn5-1* could result from the activity of other HATs recruited at the *ERF1* and *EBF2* loci. More genetic studies showed that CLV/GCN5 requires EIN3 activity to modulate ethylene-induced gene expression, but they do not regulate EIN3 expression. ChIP experiments revealed that some ethylene pathway genes such as *ERS1, ERF1, EBF2,* and *CTR1* are directly regulated in transcription expression by GCN5 through histone acetylation [44,45,46]. All the above indicate that the ethylene networks mediated by GCN5 and key transcription factors are intricate through multiple levels of gene/protein modifications and genetic interaction. However, whether GCN5 can also participate in the ethylene synthesis pathway by epigenetic modification has not been reported.

Furthermore, both *gcn5* and *ada2b* mutants exhibit aberrant auxin-related growth phenotypes [8,29]. In *gcn5-1*, the expression of *IAA3*, a key negative regulator of auxin signaling is decreased compared with that in the wild-type and *clv1-1* mutant plants. Genetic analysis of the *clv1-1 gcn5-1* showed that the *clv1-1* mutant enhances the *gcn5-1* effect on *IAA3* expression. In addition, 1-aminocyclopropane-1-carboxylate (ACC) treatment weakened the DR5: GFP signal accumulated in the dark-grown seedlings in all genotypes tested [47], which indicates that GCN5/CLV1 represents the crosstalk between ethylene and auxin in some developmental stages (Figure 2A), whereas the underlying mechanism is almost blank.

### 3.2. Mechanisms of GCN5 Involved in the Regulation of Salicylic Acid

Although GCN5 functions in the enhanced acetylation level of H3K14 in the promoter region of its target gene, whether GCN5 binding is systematically associated with gene activation at a genomic level is not clear. A recent study reported that both GCN5 binding and H3K14Ac level are positively correlated with gene expression through several global methodologies (ATAC-seq, ChIP-seq, and RNA-seq). Further GO enrichment analysis revealed that up-regulated genes by GCN5 are enriched in salicylic acid (SA) reaction-related categories, suggesting that GCN5 may be an SA-mediated immune regulator. The phenotypic analysis evidenced the hyposensitivity of *gcn5* mutants to the bacterial infection [48]. Additionally, an over-expressed *NahG* (encoding a bacterial enzyme that degrades SA) [49], and a *SID* mutation (defect in ISOCHORISMATE SYNTHASE 1, a key enzyme in SA biosynthesis) [50] were introduced into the *gcn5* mutant, respectively, eliminating the enhanced resistance to pathogen infection in *gcn5* mutant compared to the WT [48]. Moreover, plants *MYC2*, *DEFENSE NO DEATH2* (*DND2*), and *WRKY33*, involved in the control of SA accumulation are down-regulated in expression and show reduced H3K14Ac levels at their loci in *gcn5* mutants [51]. These results show that GCN5 plays an important role in plant immunity through the regulation of key genes of the SA pathway and homeostasis by histone acetylation (Figure 2B), providing the possibility of GCN5 application in the biotic stress tolerance of the plant.

### 3.3. Mechanisms of GCN5 Involved in Cellulose Synthesis

As sessile organisms, plant cells are surrounded by a cellulose-rich wall that is essential for plant morphogenesis and responses to certain external stimuli. Plants have evolved highly complex signaling pathways to participate in cell wall formation, maintenance, and plasticity [52,53,54,55]. The chitinase-like (CTL) protein CTL1 and its homolog CTL2 affect cellulose biosynthesis and play key roles in establishing interactions between cellulose microfibrils and hemicelluloses [56]. The *gcn5* mutant shows a decrease in cellulose content and more severe growth inhibition of root compared with the wild type under herbicide isoxaben treatment. Similar to the *ctl1* mutant, *gcn5* shows an increase in lignin deposition, cell shape blurring, epidermal cell swelling, and ectopic lignification [57]. ChIP-qPCR showed that GCN5 can bind directly to the promoter of target gene *CTL1* and regulate the levels of H3K9Ac and H3K14Ac to control the gene transcription (Figure 2C); constitutive expression of *CTL1* or wheat *TaGCN5* in the *gcn5* mutant all restored its cell wall integrity, suggesting that GCN5 participating in the synthesis of cellulose is conserved in dicots (e.g., *Arabidopsis*) and monocots (e.g., wheat) and dependent on its histone acetyltransferase activity, which provides evidence that GCN5 may take part in the plant architecture regulation.

### 3.4. Mechanisms of GCN5 in Fatty Acid Synthesis

Seed oil is an important natural resource used in food processing and utility and consists almost entirely of triacylglycerol (TAG) ester molecules that accumulate fatty acids (FAs) as the most abundant form of reduced carbon chains [58,59]. The five important FAs in seed oils are palmitic acid (C16:0), stearic acid (C18:0), oleic acid (C18:1), linoleic acid (LA, C18:2), and a-linolenic acid (ALA, C18:3). In *Arabidopsis*, the enzymes of the integral membrane fatty acid desaturase (FAD) family include FAD2, FAD3, FAD6, FAD7, and FAD8, of which both FAD2 and FAD3 are localized to the endoplasmic reticulum (ER) and responsible for desaturation [60]. The overexpression of FAD3 in *Arabidopsis* results in an increase of ALA (C18:3) levels from 19% to approximately 40% of the total seed FAs and a corresponding decrease of LA (C18:2) content in the seeds [61]. The mutation of GCN5 can reduce the ratio of ALA to LA in seed oil, and *FAD3*, *LACS2*, *LPP3*, and *PLAIIIβ* have been identified as GCN5 targets by RNA-Seq and ChIP detection. Accordingly, the ALA/LA ratio in *gcn5* mutants can be restored to the wild-type level by overexpression of *FAD3* [62]. It is worth noting that GCN5-dependent acetylation of H3K9/14 determines the expression level of *FAD3* in *Arabidopsis*, indicating that GCN5 regulates FA biosynthesis by affecting *FAD3* acetylation level and transcription (Figure 2D). These offer clues to mediate the composition and maximize the energy-efficient yield of oils through artificial epigenetic regulation on specific loci, which are always key goals in the crop industry [63,64].

In sum, GCN5 is involved in many kinds of hormone pathways and secondary metabolism processes. In these pathways, GCN5 generally acts on downstream target genes involved in metabolic pathways to modify the corresponding histone acetylation level positively or negatively. In addition, the interaction between GCN5 and CLV affects not only the growth of inflorescence meristem and stamens but also participates in the regulation of the ethylene pathway in *Arabidopsis*, which provides some messages for the mechanisms that GCN5 participates in hormone signaling and secondary metabolisms in other plant species.

## 4. Mechanisms of GCN5 Involved in Resistance to Abiotic Stresses

### 4.1. Mechanisms of GCN5 on Heat and Drought Stress Resistance

With global warming and temperature increase becoming more frequent, thermal and drought stresses have seriously affected the development, growth, reproduction, and yield of plants [65]. Many studies have shown that adverse external environment or stimuli affect histone modifications globally, and some epigenetic regulators are responsible for the chromatin adaptation in response to abiotic stresses in plants [66,67]. Furthermore, other adaptive strategies have been developed during evolution in plants, and some typical heat resistance mechanisms have been identified including the production of heat shock proteins (HSPs) to protect plants from heat stress [68,69]. GCN5 also plays important role in the heat stress response. Several key regulatory factors, such as thermal response transcription factor HEAT STRESS TRANSCRIPTION FACTOR A2 (HSFA2), HSFA3, ULTRAVIOLET HYPERSENSITIVE 6 (UVH6) are down-regulated in *gcn5* compared with the wild type under thermal stress [9]. ChIP detection further showed that GCN5 protein is enriched at promoters of *HSFA3* and *UVH6*, in turn promoting the acetylation of H3K9/K14 and the activation of *HSFA3* and *UVH6* under thermal response, indicating that GCN5 plays a key role in maintaining the heat resistance of *Arabidopsis* by regulating key transcription factors (Figure 3A). Moreover, the ectopic expression of wheat *TaGCN5* also rebuilt heat tolerance in *Arabidopsis gcn5* mutant [9], proposing that GCN5-mediated thermotolerance may be conserved in the flowering plant including dicots and monocots. In poplar, an important factor AREB1 was identified to recruit and enable GCN5-ADA2b mediating histone acetylation to be involved in the drought tolerance, in which the drought-responsive genes *PtrNAC006*, *PtrNAC007*, and *PtrNAC120* function downstream of AREB1-ADA2b-GCN5 ternary complex to regulate the response to drought stress [70], further supporting the conservative role of GCN5 in different abiotic stresses tolerance of the plant.

### 4.2. Effect and Mechanism of GCN5 on Cold Stress Tolerance

Low temperature causes great harm to the growth and development of plants; therefore, it is necessary to explore the underlying mechanisms of plants in response to low temperature, especially to uncover the key genes in the process of adaptation to cold stress [71,72]. GCN5 exerts roles not only in resisting heat stress but also in interacting with cold-induced transcription factors. *Arabidopsis* C-REPEAT/DRE BINDING FACTOR 1 (CBF1) protein is a key transcriptional activator that binds to the DRE/CRT regulatory element and induces *COR* (cold-regulated) gene expression to increase plant freezing tolerance [73]. In response to low temperature, *CBFs* are up-regulated in both *gcn5* and *ada2b* mutants similar to that in wild-type, but the transcription of *COR* genes is reduced in both mutants [8]. Some research indicated that CBF1 might recruit the GCN5 and AdA2b containing SAGA-like complexes to the promoters of its target genes to facilitate gene transcription. Protein interaction studies further showed that the DNA binding domain rather than transcriptional activation domain of CBF1 can directly bind with ADA2 protein [10]. Thus, the pathway involved by CBFs seems dependent on GCN5 or AdA2b partially in the cold stress response (Figure 3B). Interestingly, normal *ada2b-1* (but not *gcn5-1*) plants show more resistant to freezing than wild-type plants [10], indicating that ADA2b may depress a freezing tolerance mechanism independent of *CBF* or *COR* genes directly or indirectly. Therefore, ADA2b and GCN5 may participate in different pathways to regulate diverse plants’ stress tolerance caused by low or freezing temperatures. These results suggest that it would be very valuable to explore the underlying mechanisms involved by different histone acetylation complexes in response to the low temperature of varying degrees.

### 4.3. Mechanisms of GCN5 in Salt Stress Tolerance Reactions of Plants

Excessive soluble salt in the soil is harmful to the growth and development of plants by producing toxic ions, osmotic stress, and oxidative stress on plant cells [74,75]. Cell walls, the outermost cell structure, are essential for plant morphogenesis and response to certain external stimuli [52,53,54,55]. In recent years, some studies have shown that plant cell walls participate in the perception and response to salt stress and maintaining cell wall integrity is crucial for stress tolerance and cell protection [76]. It was found that the mRNA level of *GCN5* is increased under salt stress; *gcn5* mutant shows shortened root length, severe growth inhibition, and cell wall integrity defects compared with the wild type in response to salt stress. Combined with RNA-seq and ChIP analysis, it was found that GCN5-mediated H3K9 and H3K14 acetylation is associated with *CTL1, POLYGALACTURONASE INVOLVED IN EXPANSION3 (PGX3)*, and *MYB54* activation under salt stress, and the cellulose gene *CTL1* as a direct target of GCN5 plays important roles in cell wall integrity and salt tolerance [57] (Figure 3C). In summary, GCN5 maintains cell wall integrity and is involved in salt tolerance by regulating some cellulose synthesis genes directly. Whether GCN5 can influence other pathways or genes to regulate the cell osmotic stress still need more work to reveal.

### 4.4. Role and Mechanism of GCN5 in Mediating Iron Homeostasis in Plant

Iron is an important micronutrient and affects root morphogenesis, photosynthesis, nitrogen fixation, respiration, flower color, and fertility, whose homeostasis is essential for plant development [77,78]. On the other hand, iron excess is detrimental to plant development. Therefore, to ensure that plants can obtain adequate amounts of iron from the soil while avoiding iron excess, plants have evolved a set of sophisticated mechanisms to strictly control the absorption and distribution of iron [77]. Previous studies have found that FERRIC REDUCTASE DEFECTIVE 3 (FRD3) is involved in iron absorption and mobilization. FRD3 is a multi-drug and toxin exhalation protein that can promote iron chelation to citrate and transport of iron citrate from root to root tip [79,80]. In *Arabidopsis*, GCN5 mutation can cause iron retention in roots and reprogramming expression of genes involved in iron homeostasis. After the addition of exogenous citrate, the iron retention in *gcn5* mutation was significantly attenuated. Furthermore, ChIP assay proved that five genes (*FRD3*, [*ZUSAMMEN-CA*]*-ENHANCED 2*, *EXOCYST SUBUNIT EXO70 FAMILY PROTEIN H2*, *REQUIRES HIGH BORON 1*, *CYSTEINE-RICH RLK* [*RECEPTOR-LIKE PROTEIN KINASE*] *25*) controlling iron homeostasis are the direct targets of *GCN5* [11] (Figure 3D). The acetylation of H3K9 /H3K14 of *FRD3* mediated by GCN5 determines the dynamic expression of *FRD3*. Moreover, overexpressing *FRD3* saved the iron reserve defect of *gcn5* and partially restored plant fertility. These results show that GCN5 plays a key role in *FRD3* mediating iron homeostasis to involve plant development.

### 4.5. Mechanism of GCN5 Involved in Phosphate Starvation Response

Phosphorus is an essential macronutrient in plant metabolism and plays several important roles in plant growth and development such as energy metabolism, synthesis of nucleic acid, cell membrane formation, photosynthesis, and respiration. Phosphorus deficiency is a factor limiting plant growth and productivity in 40% of the world’s arable soils [81]. Therefore, plants have evolved regulatory mechanisms to maintain phosphorus homeostasis by improving phosphorus uptake, transport, recycling, and utilization efficiency [41,82,83]. By transcriptome analysis, 888 genes potentially involved in phosphate starvation response were identified to be regulated by *GCN5*. ChIP assay indicated that a long non-coding RNA (lncRNA) *AT4* is a direct target of GCN5, and GCN5-mediating acetylation of H3K9/K14 determines the dynamic expression of *AT4*. In the absence of phosphorus, *Arabidopsis gcn5* mutants display attenuated phosphate accumulation between roots and shoots, while the constitutive expression of *AT4* in *gcn5* mutants can partially restore phosphate redistribution. Further studies evidenced that *miR399* and its target *PHOSPHATE 2* (*PHO2*) mRNA level are influenced in the regulation of *AT4* by GCN5 [12]. In summary, GCN5-mediating histone acetylation plays an important role in the regulation of phosphate starvation response through the AT4-miR399-PHO2 pathway (Figure 3E), which provides a new epigenetic mechanism representing the interaction between histone acetylation and non-coding RNA and shows the significant function of GCN5 in plant phosphorus metabolism. To sum up, GCN5 plays multiple and important roles in plants’ resistance to diverse abiotic stresses including not only cold and heat stress, but also iron stresses and others through different downstream pathways and interacting factors.

## 5. Mechanism of GCN5 Involved in MiRNA Generation

MiRNAs, encoded by specific genes in the genome, are transcribed as primary transcripts called primary miRNAs (pri-miRNAs). Many miRNAs regulate plant development by delimiting the accumulation of target transcripts that function in development. Mutants with miRNAs accumulation deficiency, such as *dicer like1* (*dcl1*), *hyponastic leaves1* (*hyl1*), *serrate* (*se*), and *argonaute1* (*ago1*), produce pleiotropic developmental abnormalities [84,85,86,87,88]. In general, histone acetylation and miRNA play opposite functions in the regulation of gene transcriptional expression. Compared with wild-type plants, most miRNAs detected in *gcn5-2* are elevated. Some miRNAs, such as *miR159*, *miR172*, and *miR399* are increased more significantly than others (e.g., *miR165*) suggesting the preferable role of GCN5 on specific miRNA types, but the underlying mechanism is unclear. The increase of most miRNAs measured is consistent with the corresponding decrease of pri-miRNAs, revealing that the maturation process of these miRNAs may be enhanced in *gcn5* mutants. The negative function of GCN5 in miRNA generation may be realized by indirectly inhibiting miRNA mechanism genes, such as *DCL1*, *HYL1*, and *AGO1* [20]. ChIP analysis revealed that GCN5 targets a subset of miRNA genes and regulates acetylation of the H3K14 at these loci (Figure 3F). These findings suggest that *Arabidopsis* GCN5 interferes with miRNA pathways at both transcriptional and post-transcriptional levels, and histone acetylation is an epigenetic mechanism involved in regulating miRNA production. However, the underlying mechanisms of GCN5 interacting with other epigenetic modifications (e.g., histone methylation, DNA methylation, protein ubiquitination, and protein phosphorylation) in miRNA generation is not clear up to now.

## 6. Global Analysis of Binding Sites and Gene Expression Associated with GCN5 by Genome-Wide Approaches

Given the general effects of GCN5 in plant development, it is worth investigating the genome-wide binding sites and gene expression profiles mediated by GCN5. In *Arabidopsis*, the systematic analysis of *Arabidopsis* promoters (SAP) repertoire was constructed, bringing a total set of 22, 260 in silico amplicons (http://www.psb.ugent.be/SAP/ (accessed on 24 February 2021). Through ChIP-chip assay, an antibody against the GCN5 was used to detect the associated chromatin fragments in wild-type and *gcn5* mutants lacking the bromodomain, respectively. The results showed that 40% of the tested promoters were associated with GCN5 and most of which did not require the effective GCN5 bromodomain to bind. However, the bromodomain was necessary for binding to 11% of the promoter regions and related with acetylation of H3K14 in these promoters. Integrated analysis of ChIP-chip and transcriptome implied that binding of GCN5 does not show a strict correlation with gene transcription activation [14,22]. To explore the underlying mechanism between gene transcription and H3K14Ac regulated by GCN5, another ChIP-seq combined with the MNase-seq and ATAC-seq methods were applied, which revealed that GCN5 binding displays two peaks before and after the TSS (transcription start sites). Further analysis showed that one type of target genes of GCN5 displays a positive correlation between H3K14Ac on the 5′ end of the gene and its transcriptional expression; another type shows a negative relationship between H3K14Ac on their 3′ ends and gene expression (Figure 4), which may be involved in abiotic and biotic stresses, respectively. The potential reason may be partially that GCN5 can be recruited to chromatin in two ways, one is the direct interaction between bromodomain and acetylated histones, and another is its interaction with specific transcription factors associated with G-Box consensus element (CACGTG) in promoters [9,15,48]. Even so, much work is needed to illuminate the underlying mechanisms in the future. Furthermore, the acetylation level of H3K14 in mutants *gcn5-1* and *gcn5-2* is decreased significantly at 5′ terminals but increased at 3′ terminals suggesting that GCN5 reversely controls the deposition of the marker at the 5′ and 3′ ends of the gene, which proved that GCN5 has the dual and opposite functions on regulating H3K14Ac level at the 5′ and 3′ ends of its target gene. Additionally, the genome-wide level changes of H3K14Ac in the *gcn5-1* mutant are also related to the changes of H3K9Ac at the 5′ and 3′ ends [48].

Apart from acetylation, eight types of Lysine acylations have recently been identified on histones including propionylation, butyrylation, 2-hydroxyisobutyrylation, succinylation, malonylation, glutarylation, crotonylation, and β-hydroxybutyrylation which affect gene expression and are structurally and functionally different from the histone Lys acetylation [89]. A structural study of human GCN5 [90] in complex with some acyl-CoAs revealed some clues behind the abilities of catalyzing acylation reactions. However, correlation and molecular mechanisms between GCN5 and other acylations are almost blank in the plant. These results also forward us to reveal a more global analysis of Lysine acetylation types, binding sites pattern, and the gene expression profile for GCN5 in other species.

## 7. Discussion

GCN5 has been intensely studied to prove that it is a general histone acetylation modifier involved in diverse eukaryote development and response to adverse conditions. But, even so, the emerging evidence is exploring the much more novel functions and mechanisms of GCN5. This review summarizes the updated roles and underlying mechanisms of GCN5 in the development of floral organs, epidermis, organogenesis, and resistance to stresses as well as some hormonal and secondary metabolism pathways in the plant (Table 1 and Table 2). More interesting, the intensive studies for mechanisms showed that GCN5 can bind to many protein molecules and may be involved in a variety of regulatory processes. Some complexes associated with GCN5 show multifunctional features. For example, the interaction between GCN5 and CLV1 regulates not only root and shoot meristem structure but also crosstalk of ethylene and auxin signaling pathways, which offers clues and messages of the interaction between GCN5 and other phytohormones. Another conserved interacting factor of GCN5 is ADA2b. The combination of GCN5 and ADA2b facilitates the SAGA-complex formation and function, and defect in this process results in plant dwarfism and a series of leaf and flower phenotypic abnormalities. The ADA2b protein can bind directly to GCN5 through its N-terminal domain, or to the cap domain of GCN5 through a previously unmapped region in the middle of the ADA2 protein. ADA2 can enhance the GCN5 acetylation ability of histone in vitro. In turn, ADA2 protein can be acetylated by GCN5, and the motif of ADA2 protein is unique in the plants, but not in fungi and animals [8], implying that ADA2 differentiate functionally during evolution and may play specific and important role in the SAGA complex of plant. Furthermore, a special model of GCN5-TAF1-HD1 in Arabidopsis photomorphogenesis is proposed here. The three enzymatic proteins can involve different complexes to regulate the same target genes such as *CAB2*, *RBCS1A*, and *IAA3* to control the photomorphogenesis; the key factor HY5 play some bridge role in this network, which indicates the key roles of histone acetylation and some hints for the photomorphogenesis in the terrestrial plant, whereas the delicate underlying mechanisms are unclear.

Additionally, alternative splicing in *GCN5* transcripts have been reported in humans as well as *Brachypodium distachyon* (i.e., *GCN5*, *L-GCN5*, and *S-GCN5* splice variants) [92], in which GCN5 and L-GCN5 but not S-GCN5 could interact with ADA2, indicating that these isoforms function in the diversification of the SAGA complex association, and shed light on the differences in the protein organization of plant GCN5-containing complexes [92]. Thus, plants may hold compositionally distinct SAGA-like complexes based on the different histone acetyltransferase subunits. The recent work showed that three functionally redundant paralogs, SCS1, SCS2A, and SCS2B (SCS1/2A/2B), and a TAF-like subunit, are required for the function of the SAGA complex in histone acetylation associated with GCN5 and ADA2b, contributing to the illumination of the SAGA complex characteristic [93]. All these indicate that further investigation to explore the specific components of different SAGA complex associated with GCN5 or ADA2 isoforms is more important and meaningful; on the basis of this, we may understand the mechanical foundation and molecular mechanism among various proteins and GCN5 involved in different biochemical pathways and physiological processes in the plant. From above, the intricate interaction between GCN5 and ADA2b shows us a multidimensional model of protein-protein interaction and indicates the potential new mechanisms of GCN5 in gene and protein regulation. In common, both histone acetylation by GCN5 and transcription factor binding to *cis*-elements facilitate the locus transcription and gene activation, however, which one is the leading activator is unknown to the specific gene regulation. The application of some advanced technologies such as high-resolution Electron Cryomicroscopy and 3D image capture would be beneficial to explore the bona fide machinery of SAGA complex associated with GCN5 and underlying mechanism. HATs and HDACs are responsible for two types of reaction to balance the histone acetylation, respectively, however, whether the direct interaction between these two kinds of enzymatic complexes occurs in the cell is unclear, and associated research needs to be reinforced. These studies indicate that the investigation of other unknown interacting factors of GCN5 would contribute to the understanding of the GCN5 network in cellular and molecular levels of eukaryotes development.

To date, proteome and acetylome analyses have explored massive non-histone proteins acetylation in mammalian cells. Acetylation on non-histone proteins affects protein functions through regulating protein stability, enzymatic activity, subcellular localization, and crosstalk with other post-translational modifications [94] In *Arabidopsis*, a study revealed that HDA14 is an organellar-localized HDA1 class protein in the chloroplasts and can target and regulate RuBisCO activity through the deacetylation at Lysine 438 of RuBisCO [95]. These results promote us to explore whether GCN5 also modifies non-histone proteins to affect protein function. With specific mutants like *gcn5* and acetylome analysis, this issue could be partially explained.

In recent years, CRISPR-Cas nuclease-directed plant genome editing has become a promising method to change gene function as required and create new plant varieties. Taking advantage of this system and the characteristic of GCN5 activating gene expression by chromatin modification, a CRISPR activation (CRISPRa) system, expressing chimeric dCas9HAT machinery was established, and the associated stable transgenic plants were first generated. The results showed that CRISPRa dCas9^HAT^ mechanism increased the promoter activity of the interest gene. Therefore, dCAS9 and HAT fusion (dcas9HAT) combined with directed targeted sgRNAs seems to be the potential to actively regulate the activity of targeted promoters and genes [96], which provides a new way for plants to improve desired characteristics and cope with diverse stresses with histone acetyltransferases such as GCN5.

In this paper, we focus more on how GCN5 regulates plant growth and development as well as stress response during the process of acetylation of H3K9/K14 in *Arabidopsis*. GCN5 exhibits versatile roles in the plant through interacting with and regulating key factors, some metabolisms, or phytohormone pathways. Moreover, we should extend our focus from the model plant to other crops such as cotton, maize, wheat for protein acetylation study, and investigate more GCN5 homologs in other plant species. More work should focus on the internal connection and interaction factors of GCN5 in the development of different plant organs (e.g., trichome, seed) to illuminate the complete network of GCN5 in the plant. The further functional characterization of the novel interaction factors, substrates as well as regulators will pave the way for a deeper understanding of molecular mechanisms of GCN5 in eukaryote development.

## Figures and Tables

**Figure 1 cells-10-00979-f001:**
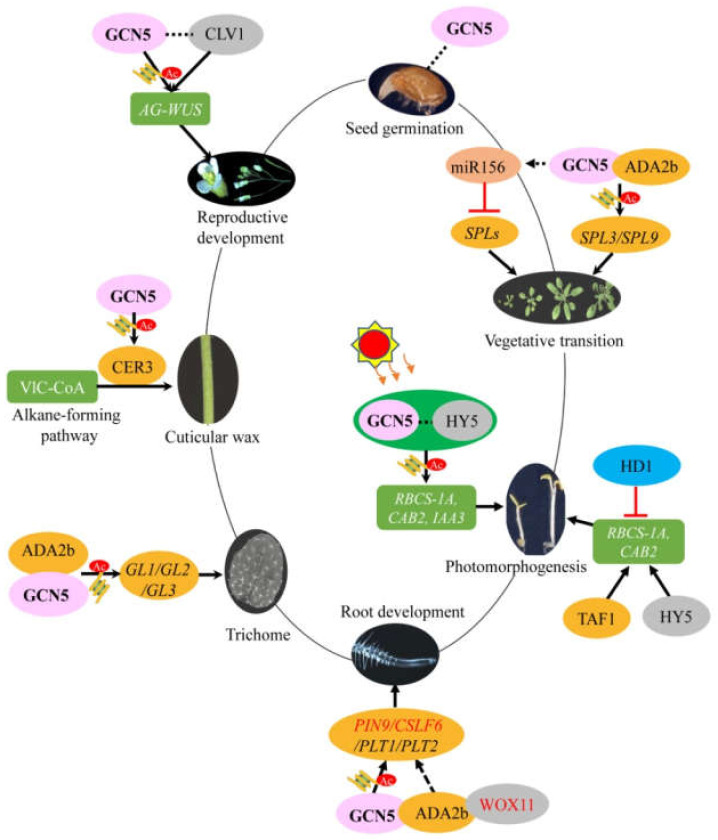
The roles and mechanisms of GCN5 in plant growth and development. In *Arabidopsis*, GCN5 plays different roles in the whole life cycle through different pathways. First, GCN5 interacts with ADA2b as a complex to regulate *SPL3/SPL9* directly through histone acetylation to involve the juvenile-to-adult vegetative phase, which is independent of the pathway of *miRNA156*-*SPLs* action. Interestingly, GCN5 also regulates *pri-miR156a* expression positively. Then, a module made of GCN5, TAF1, and HD1 contributes to the photomorphogenesis and vegetative development of plants through delicate histone acetylation regulation, in which GCN5 and TAF1 function synergistically, and HD1 functions oppositely with them. HY5, the key photomorphogenesis factor is responsible for the recruitment of GCN5 and TAF1 in different ways. GCN5 interacts with HY5 genetically and functions in the same way in morphogenesis regulation. While TAF1 functions synergistically with HY5. *RBCS-1A*, *CAB2*, and *IAA3* play different roles as target genes of histone acetyltransferase and histone deacetylase in this pathway. For root meristem development, GCN5, together with ADA2b, can increase transcripts of *PLT1* and *PLT2* by histone acetylation regulation to adjust the stem cells meristem, furthermore, ADA2b also functions independently of GCN5 to affect the stem cell niche maintenance. In rice, WOX11 can interact with ADA2b to recruit the GCN5 associated histone acetyltransferase complex and together regulate downstream *PIN9*, *CSLF6*, and other genes to facilitate the crown root meristem development. In trichome development, the GCN5-ADA2b complex regulates core genes *GL1*, *GL2,* and *GL3* through histone acetylation modifications to mediate trichome initiation and branching. In the stem cuticular wax formation, GCN5 regulates the *ECERIFERUM3* transcription by histone acetylation to influence cuticle membrane and wax biosynthesis. Considering the flowering, GCN5 and CLAVATA 1 regulate AG-WUS through direct histone acetylation modification to involve several floral organ developments synergistically, but the relationship between GCN5 and CLAVATA 1 is still unclear. However, it is unknown whether GCN5 affects seed development or germination associated traits. The images on the circle represent the different organs and developmental stages of *Arabidopsis thaliana*. GCN5 is indicated in the pink oval. The black arrows indicate active regulation, and the red bars indicate inhibition. The straight line represents the direct interaction, and the dashed line represents the indirect action. The histone acetylation modifications are represented with a graphic histone binding with the acetyl group (a red circle).

**Figure 2 cells-10-00979-f002:**
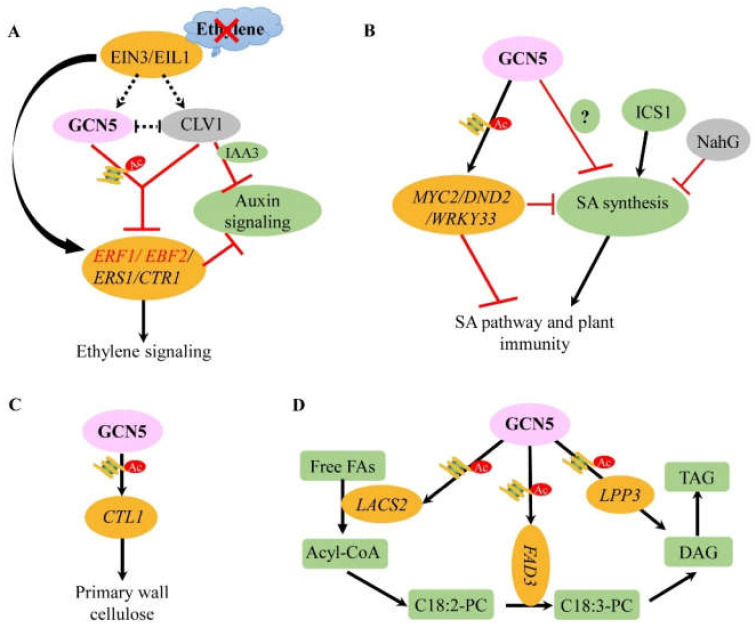
The mechanisms of GCN5 in plant hormone biosynthesis and secondary metabolic pathways. (**A**) When ethylene is absent, GCN5 and CLV1 are involved in ethylene signaling through regulating some key genes transcription (e.g., *ERS1*, *ERF1*, *EBF2*, *CTR1*) by histone acetylation, which is dependent on the EIN3 factor; meanwhile, GCN5 and CLV1 also mediate *IAA3* and auxin signaling synergistically, proposing the crosslink between ethylene and auxin. But GCN5 and CLV1 show antagonistic actions on the histone acetylation of H3K9/14, which results in the up-regulation of *ERS1*, *ERF1*, *EBF2*, *CTR1* in the *clv1-1 gcn5-1* double mutant. Moreover, EIN3 can directly bind the promoters of *ERF1* and *EBF2* to control their transcription too (black curved arrow). (**B**) GCN5 regulates downstream targets *MYC2*, *DND2*, and *WRKY33* expression through histone acetylation, which inhibits the SA synthesis and accumulation; on the other hand, GCN5 mediates SA synthesis through an unidentified pathway independent of NahG and ICS1 to participate in SA-mediated plant immunity. (**C**) GCN5 can mediate the histone acetylation level of *GTL1* to affect its transcription and the associated cellulose synthesis. (**D**) In fatty acid synthesis, GCN5 can directly regulate histone acetylation of *FAD3* and others (e.g., *LACS2*, *LPP3*) to mediate their transcription expression, and involve the different steps of fatty acid synthesis and accumulation. DAG, diacyl glycerol; TAG, triglyceride. GCN5 is indicated in the pink oval. The black arrows indicate active regulation, and the red bars indicate inhibition. The straight line represents the direct action, and the dashed line represents the indirect action. The histone acetylation modifications are represented with a graphic histone binding with the acetyl group (a red circle).

**Figure 3 cells-10-00979-f003:**
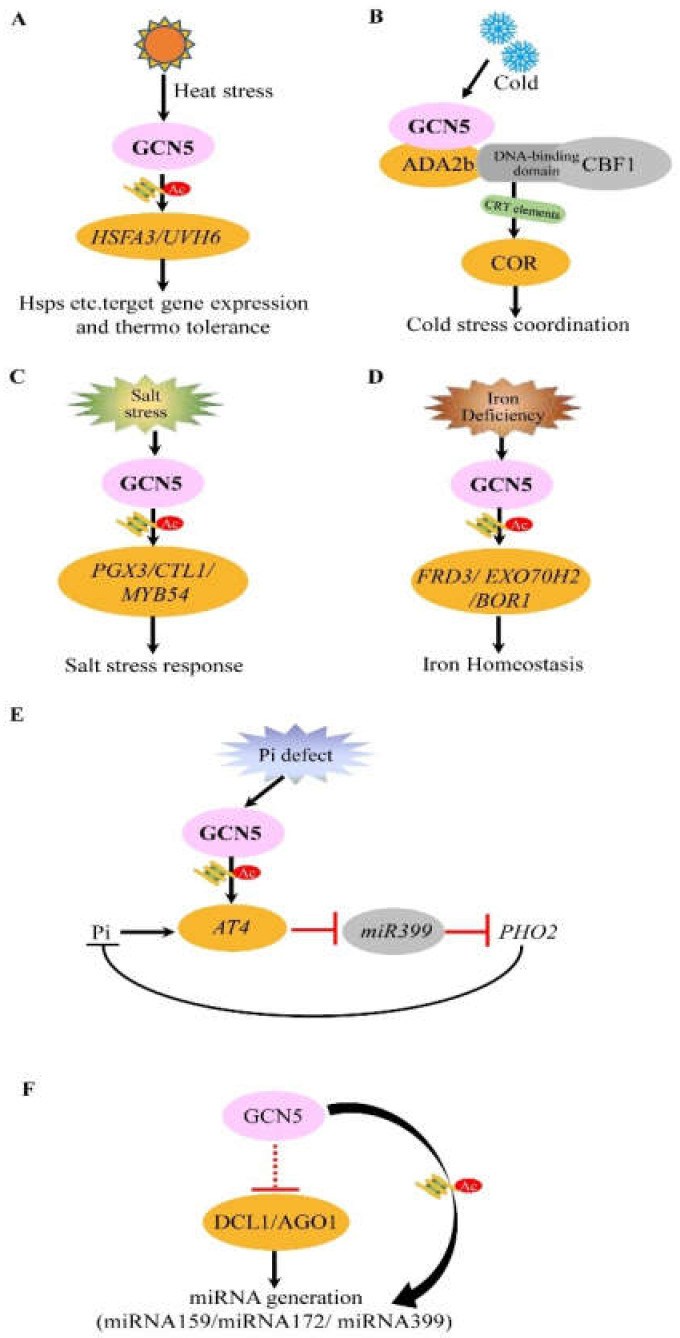
Underlying mechanisms of GCN5 in response to abiotic stresses and miRNA generation. (**A**) In response to heat stress, GCN5 directly functions on two key transcriptional factors HSFA3 and UVH6 encoding genes by histone acetylation modifications to activate their transcription, in turn, activating some heat shock protein functions and mediating plant heat resistance. (**B**) Encountering cold stress, CBFs recruit GCN5 and ADA2b through the DNA binding domain to activate its expression, then bind the *CRT* elements in the key genes *COR* promoter and promote transcription and increase the plant resistance to low-temperature stresses. However, the detailed mechanisms of interaction among GCN5, ADA2b, and CBFs are unclear. (**C**) In response to salt stress, GCN5 is up-regulated to promote the expression of downstream genes of *PGX3*, *CTL1*, *MYB54* through histone acetylation, in turn, to increase salt stress tolerance. (**D**) In iron deficiency, GCN5 regulates directly the histone acetylation of *FRD3*, *EXO70H2*, and *BOR1* to promote their expression, which in turn facilitates synthesis, transport, and homeostasis maintenance of iron in the cell. (**E**) In response to phosphate starvation, a long non-coding RNA (lncRNA) *AT4* is up-regulated and identified as a target of GCN5 through histone H3 acetylation modifications, then downstream *miR399* and its target *PHOSPHATE2* is repressed and promoted respectively to mediate phosphate proper distribution. (**F**) GCN5 is involved in the miRNA production by regulating the miRNA machinery *AGO1* and *DCL1* indirectly. Further, it can also regulate some miRNA genes directly by histone acetylation modifications. GCN5 is indicated in the pink oval. The black arrows indicate active regulation, and the red bars indicate inhibition. The straight line represents the direct action, and the dashed line represents the indirect action. The histone acetylation modifications are represented with a graphic histone binding with the acetyl group (a red circle).

**Figure 4 cells-10-00979-f004:**
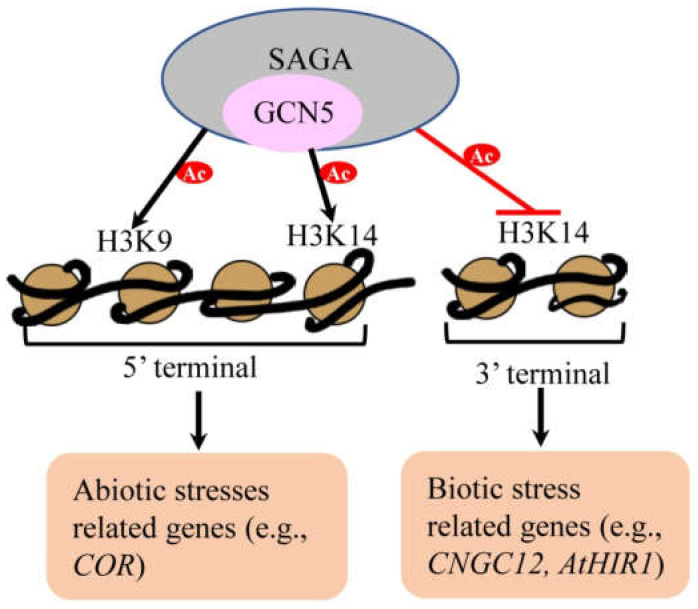
The molecular mechanisms of GCN5 involved in abiotic and biotic stresses on the genomic level. On the genomic level, the SAGA complex associated with GCN5 exerts dual and opposite effects on H3K14Ac level at the 5′ and 3′ ends of target genes as well as a positive role on H3K9Ac. Generally, the positive (e.g., *COR*) and negative (e.g., *CNGC12*, *AtHIR1*, and *WRKY57*) regulated genes are correlated with abiotic and biotic stress responses, respectively.

**Table 1 cells-10-00979-t001:** The interaction factors of GCN5 in plant development and abiotic stress responses.

Name	Relationship to GCN5	Gene Locus	Annotation	References	Developmental Events or Pathways
ADA2b	Synergy and PPI	AT4G16420	HOMOLOG OF YEAST ADA2 2B	[8]	Constitutive and structural
TAF1	Indirect interaction	AT3G19040	TBP-ASSOCIATED FACTOR 1 with histone acetyltransferase activity	[3]	Photomorphogenesis
HD1	Indirect interaction	AT4G38130	Histone deacetylase ^1^,	[22]	
HY5	Indirect interaction	AT5G11260	transcriptional activator of light signaling transcriptional activator of light signaling transcriptional activator of light signaling transcriptional activator of light signaling ELONGATED HYPOCOTYL 5,	[14]	
CLV1	Synergy/ Indirect interaction	AT1G75820	CLAVATA1, transmembrane receptor kinase with an extracellular leucine-rich domain.	[33,47]	Shoot and root meristem
WOX11	Indirect interaction	Os07g48560	WUSCHEL-related homeobox gene family	[30]	
CBF1	Synergy and PPI	AT4G25490	C-REPEAT/DRE BINDING FACTOR ^1^, cold-induced transcription factor	[10]	Abiotic stress response
AREB1	PPI	Potri.002G125400	ABA-Responsive Element Binding	[70]	

^1^ PPI, protein-protein interaction.

**Table 2 cells-10-00979-t002:** The target genes of GCN5 in plant development and abiotic stress responses.

Gene Name	Relationship to GCN5	Gene Locus	Annotation	References	Developmental Events or Pathways
*mir156*	–	AT2G25095/ AT5G11977	Micro RNA156	[16,19]	Vegetative development and Photomorphogenesis
*SPL3*	Direct downstream target	AT2G33810	SQUAMOSA PROMOTER BINDING PROTEIN-LIKE 3	[16]	
*SPL8*	Direct downstream target	AT1G02065	SQUAMOSA PROMOTER BINDING PROTEIN-LIKE 8	[16]	
*SPL9*	Direct downstream target	AT2G42200	SQUAMOSA PROMOTER BINDING PROTEIN-LIKE 9	[16]	
*IAA3*	Direct downstream target	AT1G04240	INDOLE-3-ACETIC ACID INDUCIBLE 33	[22]	
*RBCS-1A*	Direct downstream target	AT1G67090	RIBULOSE BISPHOSPHATE CARBOXYLASE SMALL CHAIN 1A	[22]	
*AG*	Direct downstream target	AT4G18960	AGAMOUS	[32,33]	Shoot and root meristem development
*WUS*	Direct downstream target	AT2G17950	WUSCHEL	[32,33]	
*PLT2*	Direct downstream target	AT1G51190	PLETHORA2, key stem cell transcription factors	[29]	
*OsPIN9*	Direct downstream target	LOC_Os01g58860	PIN-FORMED9, encodes an auxin efflux carrier	[30]	
*OsCSLF6*	Direct downstream target	LOC_Os08g06380	CELLULOSE SYNTHASE-LIKE F6	[30]	
*Os1BGLU5*	Direct downstream target	LOC_Os01g70520	Beta-glucosidase homologue	[30]	
*GL1*	Direct downstream target	AT3G27920	GLABRA1, core trichome initiation regulator genes	[37]	Trichome development
*GL2*	Direct downstream target	AT1G79840	GLABRA2, core trichome initiation regulator genes	[37]	
*GL3*	Direct downstream target	AT5G41315	GLABRA3, core trichome initiation regulator genes	[37]	
*DCL1*	Direct downstream target	AT1G01040	DICER-LIKE ^1^, miRNA maturation	[20,84]	MiRNA generation and regulation
*SE*	Direct downstream target	AT1G27100	Zinc-finger domain protein SERRATE	[20,85]	
*HYL1*	Direct downstream target	AT1G09700	HYPONASTIC LEAVES 1	[20,88]	
*AGO1*	Direct downstream target	AT1G48410	ARGONAUTE 1, an RNA Slicer that selectively recruits microRNAs and siRNA	[20,86]	
*CTL1*	Direct downstream target	AT1G05850	CHITINASE-LIKE ^1^ gene,	[56]	Lipid metabolism, cell wall and oil composition
*FAD3*	Direct downstream target	AT2G29980	FATTY ACID DESATURASE 3	[62]	
*CER3*	Direct downstream target	AT5G57800	ECERIFERUM3	[41]	
*LACS2*	Direct downstream target	AT1G49430	LONG-CHAIN ACYL-COA SYNTHETASE 2	[62]	
*LPP3*	Direct downstream target	AT3G02600	LIPID PHOSPHATE PHOSPHATASE 3	[62]	
*PLAIIIb*	Direct downstream target	AT3G54950	PATATIN-RELATED PHOSPHOLIPASE IIIBETA	[62]	
*EIN3*	Direct downstream target	AT3G20770	ETHYLENE INSENSITIVE 3	[47]	Phytohormone and abiotic stresses response
*miR399*	Indirect	AT2G34202	Stress response	[12]	
*HSFA3*	Direct downstream target	AT5G03720	Heat Stress Transcription Factor3	[9]	
*UVH6*	Direct downstream target	AT1G03190	ULTRAVIOLET HYPERSENSITIVE 6	[9]	
*MYB54*	Direct downstream target	AT1G73410	MYB DOMAIN PROTEIN 54	[57]	
*PGX3*	Direct downstream target	AT1G48100	POLYGALACTURONASE INVOLVED IN EXPANSION3	[57]	
*AT4*	Direct downstream target	AT5G03545	Long non-coding RNA, response to phosphate starvation	[12,91]	
*FRD3*	Direct downstream target	AT3G08040	FERRIC REDUCTASE DEFECTIVE3	[62]	
*PtrNAC006*	Direct downstream target	Potri.002G081000	NAC (NAM, no apical meristem) DOMAIN CONTAINING PROTEIN 006	[70]	
*PtrNAC007*	Direct downstream target	Potri.007G099400	NAC DOMAIN CONTAINING PROTEIN 007	[70]	
*PtrNAC120*	Direct downstream target	Potri.001G404100	NAC DOMAIN CONTAINING PROTEIN 120	[70]	

^1^ indicating the correlation with GCN5 is unclear; PPI, protein-protein interaction.

## Data Availability

Not applicable.

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
