# Peer review of "Updated Mechanisms of GCN5—The Monkey King of the Plant Kingdom in Plant Development and Resistance to Abiotic Stresses"

_cells, 2021, doi:10.3390/cells10050979_

Round 1

Reviewer 1 Report

The manuscript by Gan et al. summarizes what is known about the Histone acetyltransferase GCN5 in plants. Although this is an interesting review, it requires a substantial amount of editing. Some of the figures are misleading. Several references are not cited correctly. 

For instance,

ln 62, 63 delete "development."

ln 70, MMC1 is not a HAT

ln 81, delete "well."

Ln 101-103 "Some genetic....to epistatic....transition". This statement is incorrect. The effect of GCN5 on SPLs are independent of microRNA function.

ln 141, replace "photo" with light

ln 177, please rephrase it 

ln 213-215, there is no evidence that the CLV1-GCN5 regulate AG; the reference [33] does not report this statement.

ln 250 and 255, reference 38 is the same as 39.

ln 303 "the expression levels of ERF1 and EBF2 were slightly increased after ethylene treatment in clv1-1/gcn5-1 [44-46]". Wrong references...

Ln 310-312 incorrect references. Moreover, the effect of GCN5 in ERF1 and EBF2 is found in the absence of ethylene. Suggesting Figure 2A is misleading; EIN3-EIL1 is required for GCN5-CLV1 synergistic action.

Ln 626 "GCN5 binding with CLV1 regulates not only root and shoot meristem...". There is no evidence that GCN5 binds to CLV1.....

Please write the Discussion from the beginning by highlighting the unique features of this review instead of repeating the results. The two tables are redundant with the current knowledge, and there are at least 3 microarrays and RNA seq experiment coupled with ChIP that suggest many more GCN5 target genes. 

Reviewer 2 Report

This is a well written review article. The topics is an interesting one, the references are up to date, illustration figures are well placed, summaries and conclusions are reasonable.

Author Response

Thank you for the approval and the comments.

Reviewer 3 Report

Authors show a thorough review of the molecular role of the GCN5 protein on multiple processes of plant development and defense against biotic and abiotic stresses. The manuscript is well written and comprehensive and I just suggest minor modifications. 

  •    Figure 3 is difficult to read. I suggest to enlarge it.
  •    Authors explain that GCN5 affect plant hormone homeostasis specially to ethylene and salicylic acid. The biosynthesis of these hormones is intimately associated to jasmonate biosynthesis (JA). Since, GCN5 also affects LA (18:3) synthesis (the precursor of JA), I wonder if there is any knowledge about the effect of GCN5 on the JA pathway. Specially when is reported that aos mutation in Arabidopsis repress the expression of the GCN5-related N-acetyltransferase (GNAT) gene.
  • Authors should be cautious when write about thermal stresses. In the heat stress resistance, they mix high temperature and drought stress. These two stresses used to act in combination but not necessarily, so the role of GCN5 on one stress or other or a combination of both must be clarified. Related to cold stress is also important to clarify whether the role of GCN5 is related to low or freezing temperatures.

Round 2

Reviewer 1 Report

The authors respond to most of my previous request; therefore I endorse the manuscript for publication.